# Intragenic Deletions in *FLNB* Are Part of the Mutational Spectrum Causing Spondylocarpotarsal Synostosis Syndrome

**DOI:** 10.3390/genes12040528

**Published:** 2021-04-05

**Authors:** Kaya Fukushima, Padmini Parthasarathy, Emma M. Wade, Tim Morgan, Kalpana Gowrishankar, David M. Markie, Stephen P. Robertson

**Affiliations:** 1Department of Women’s and Children’s Health, Dunedin School of Medicine, University of Otago, Dunedin 9016, New Zealand; kayfu009@student.otago.ac.nz (K.F.); mini.parthasarathy@otago.ac.nz (P.P.); emma.wade@otago.ac.nz (E.M.W.); tim.morgan@otago.ac.nz (T.M.); 2Department of Medical Genetics, Apollo Hospitals, Chennai 600034, India; gskalpana@yahoo.com; 3Department of Pathology, Dunedin School of Medicine, University of Otago, Dunedin 9016, New Zealand; david.markie@otago.ac.nz

**Keywords:** filamin B, *FLNB*, spondylocarpotarsal synostosis syndrome, SCT

## Abstract

Spondylocarpotarsal synostosis syndrome (SCT) is characterized by vertebral fusions, a disproportionately short stature, and synostosis of carpal and tarsal bones. Pathogenic variants in *FLNB*, *MYH3*, and possibly in *RFLNA*, have been reported to be responsible for this condition. Here, we present two unrelated individuals presenting with features typical of SCT in which Sanger sequencing combined with whole genome sequencing identified novel, homozygous intragenic deletions in *FLNB* (c.1346-1372_1941+389del and c.3127-353_4223-1836del). Both deletions remove several consecutive exons and are predicted to result in a frameshift. To our knowledge, this is the first time that large structural variants in *FLNB* have been reported in SCT, and thus our findings add to the classes of variation that can lead to this disorder. These cases highlight the need for copy number sensitive methods to be utilized in order to be comprehensive in the search for a molecular diagnosis in individuals with a clinical diagnosis of SCT.

## 1. Introduction

Spondylocarpotarsal synostosis syndrome (SCT; MIM# 272460) is a skeletal dysplasia characterized by vertebral fusions, a disproportionately short stature, and synostosis of carpal and tarsal bones. It exhibits both locus and allelic heterogeneity, with pathogenic variants in *FLNB*, *MYH3*, and the more recently proposed *RFLNA*, being reported to cause this condition [1,2,3].

To date, 16 different families have been reported to have *FLNB*-related SCT [1,4,5,6,7,8,9]. Affected individuals have homozygous or compound heterozygous truncating variants in *FLNB*. All of the reported variants are single nucleotide variants (SNVs) or small insertions or deletions (indels) that are predicted to result in a truncation of protein translation. Several of these variants have been confirmed to result in nonsense mediated decay [5,9], confirming that it is the absence of the encoded filamin B protein that causes SCT. In contrast, SNVs and in-frame indels that are predicted to have a gain-of-function effect are responsible for a number of autosomal dominant skeletal dysplasias including Larsen syndrome (MIM# 150250), atelosteogenesis type I (MIM# 108720) and type III (MIM# 108721), and boomerang dysplasia (MIM# 112310) [1,10].

In the present study, we describe two further families with phenotypic features typical of SCT. Affected individuals were found to be homozygous for novel intragenic deletions in *FLNB* predicted to result in a frameshift thus contributing to the evidence that SCT is caused by the loss-of-function of *FLNB*. Our findings expand the molecular spectrum of SCT and highlight the importance of using copy number sensitive methods when screening for mutations in this condition.

## 2. Materials and Methods

### 2.1. Subjects and Ethical Consent

Two individuals affected by SCT were identified through physician-initiated referral and consented to participate in a research study under approved protocols MEC/08/08/094 and 13/STH/56 (Health and Disability Ethics Committee, New Zealand).

### 2.2. Sequence Analysis

Genomic DNA from affected individuals and their families were received from the referring laboratories. The entire coding sequence of *FLNB* was PCR amplified from the DNA of the affected individuals using AmpliTaq Gold DNA Polymerase (Applied Biosystems, Foster City, CA, USA). These fragments were sequenced with BigDye Terminator v3.1 (Applied Biosystems) on the 3730*xl* DNA Analyzer (Applied Biosystems).

Whole genome sequencing (WGS) was performed on DNA from the affected individuals and analyzed for SNV/indels and larger structural variants as previously described [11]. Briefly, DNA libraries were prepared using the TruSeq Nano DNA Library Prep kit v2.5 (Illumina, San Diego, CA, USA) to generate paired-end 150 base pair (bp) reads. Reads were aligned to the reference sequence (GRCh37 assembly) using the Burrows-Wheeler Aligner v0.7.17 with the MEM algorithm [12]. GATK HaplotypeCaller v3.8 and Manta v1.4.0 were used to call SNV/indels and larger structural variants respectively. Variants were visually inspected using Integrative Genomics Viewer (IGV) [13]. Runs of homozygosity (ROH) were identified using the ROH tool from BCFtools v1.11 [14]. A co-efficient of inbreeding (F_ROH_) was calculated for probands, using a minimum length threshold of 1.5 Mb for ROH segments and a hg19 genome length estimate of 2881 Mb, to allow direct comparison to values from a recent study in a large population [15]. The identified variants likely to be responsible for SCT in these individuals were uploaded to ClinVar.

Genotyping of family members was conducted using allele-specific PCR employing two pairs of primers for each deletion (Figure 1b,f). The first pair (pair 1) was designed to have one primer complementary to a region within the deletion so that amplification was only possible from the wild-type (WT) allele. The second pair (pair 2) was designed to flank the deletion so that amplification would occur only from the deletion-containing allele since a product from the WT allele would be too large for the cycling conditions used. For family A, pair 1 was designed to produce a 933 bp product and pair 2 a 262 bp product. For family B, pair 1 was designed to produce a 444 bp product, and pair 2 a 178 bp product. The reactions were conducted using AmpliTaq Gold DNA Polymerase (Applied Biosystems). Primer sequences are presented in Appendix A and a detailed figure illustrating the design of this experiment can be found in Appendix A. PCR conditions are available upon request.

## 3. Clinical Reports

### 3.1. Family A

Proband A is a 31-year-old female, born to healthy parents who were from a population with known high rates of consanguinity although explicit details about the parents are unknown. She presented with short stature with a height of 144 cm (−2.4 SD) due to a short trunk and significant scoliosis, and additionally had flat feet. She was of normal intelligence and had mild bilateral sensorineural deafness.

Radiographic examination at 6 years demonstrated fusions between T3 and T11 vertebrae (Figure 1a). In the hands, capitate–hamate and scaphoid-trapezium fusions were observed bilaterally. In addition, the feet showed fusions between the talus and navicular bones and between the talus and the calcaneus on the right.

She has had multiple surgeries for her vertebral problems as well as surgery to stabilize her patella following repeated subluxations.

### 3.2. Family B

Proband B is a 9-year-old male, a child of healthy consanguineous parents, who presented with a short trunk and clinically normal hands and feet. He had mixed conductive and sensorineural hearing loss and was non-dysmorphic.

Radiographic examination at 9 years demonstrated the fusion of T6-T9 vertebrae and bilateral capitate–hamate fusion but no abnormalities in the feet were detected (Figure 1e).

## 4. Results

Following the clinical diagnosis of SCT, Sanger sequencing of the entire coding region of *FLNB* was undertaken for both affected individuals. In both cases, there was an inability to PCR amplify several consecutive exons and this was considered to be suggestive of the presence of homozygous intragenic deletions. Subsequently WGS was pursued to confirm this and determine the precise boundaries of the deletions in each individual.

Proband A was found to be homozygous for a 6432 bp deletion (NM_001457.3:c.1346-1372_1941+389del, NC_000003.11:g.58086558_58092989del, ClinVar accession: SCV001482266) which removes exons 9-12 of *FLNB* and results in a frameshift (Figure 1b). This deletion was situated within a 40 Mb autozygous region (NC_000003.11:g.20957810-61653515), as indicated by ROH analysis. The calculated F_ROH_ was a rather unremarkable 0.019, with the majority of this value contributed by the single large segment of ROH around *FLNB*. Possible alternative explanations for apparent homozygosity of the intragenic *FLNB* deletion, other than inheritance from a common ancestor were considered, such as a large encompassing deletion inherited from one parent or segmental uniparental isodisomy [16]. However, allele-specific PCR demonstrated that both parents were heterozygous for this deletion (Figure 1c,d), which was consistent with the recessive mode of inheritance observed in *FLNB*-related SCT. No other variants of explanatory significance were found on WGS.

In proband B, a homozygous 6166bp deletion (NM_001457.3:c.3127-353_4223-1836del, NC_000003.11:g.58108467_58114632del, ClinVar accession: SCV001482267) encompassing exons 21-24 of *FLNB*, also resulting in a frameshift was identified. This individual was similarly demonstrated to be autozygous in the region housing this variant (NC_000003.11:g.22163190-59738749). The calculated F_ROH_ of the proband was 0.173, suggesting significant relatedness between the parents. The father was heterozygous for this deletion, and while the mother’s DNA was not available, the maternal grandmother was heterozygous on allele-specific PCR (Figure 1g,h). This was consistent with this deletion being inherited. No other variants of explanatory significance were found on WGS.

## 5. Discussion

SCT was first identified to be caused by biallelic truncating variants in *FLNB* by Krakow et al. in 2004 [1]. Reported variants have all been SNVs or small indels that result in nonsense or frameshifting effects, including splice site variants that remove out-of-frame exons [9]. To our knowledge, this is the first time that large intragenic deletions in *FLNB* have been reported in individuals with SCT.

Both deletions reported here involve the removal of several exons and result in a frameshift. They are expected to lead to the loss of expression of the filamin B protein through nonsense-mediated transcript decay, but appropriate samples were not available to confirm this. A reduction in filamin B at either the protein or transcript level has been demonstrated in several previously reported frameshifting and nonsense variants in *FLNB* [5,9]. This, and the finding that *FLNB* null mice phenocopy SCT, including features such as vertebral fusions and carpal synostoses [5,17], point toward a pathogenesis for SCT involving the loss of filamin B.

The individuals presented here are homozygous for these deletions and this enabled the detection of these deletions through traditional PCR-based sequencing techniques. However, it is important to note that similar deletions could be missed if they were part of a compound heterozygous genotype. This risk can be mitigated by using copy number sensitive methods such as WGS or multiplex ligation-dependent probe amplification (MLPA) but not chromosomal microarray as this latter option lacks the resolution to detect all possible intragenic multi-exonic deletions. Given the locus and allelic heterogeneity of this condition, it is imperative that diagnosticians be aware of the possibility of intragenic *FLNB* mutations and consider this mutational mechanism when screening for variants in SCT to optimize the sensitivity of such analyses.

## 6. Conclusions

In conclusion, we present two further cases of SCT in which homozygous intragenic deletions in *FLNB* were identified. These cases broaden the molecular spectrum of this condition and highlight the need for copy-number sensitive methods to be used in the search for a genetic diagnosis in individuals with clinically diagnosed SCT.

## Figures and Tables

**Figure 1 genes-12-00528-f001:**
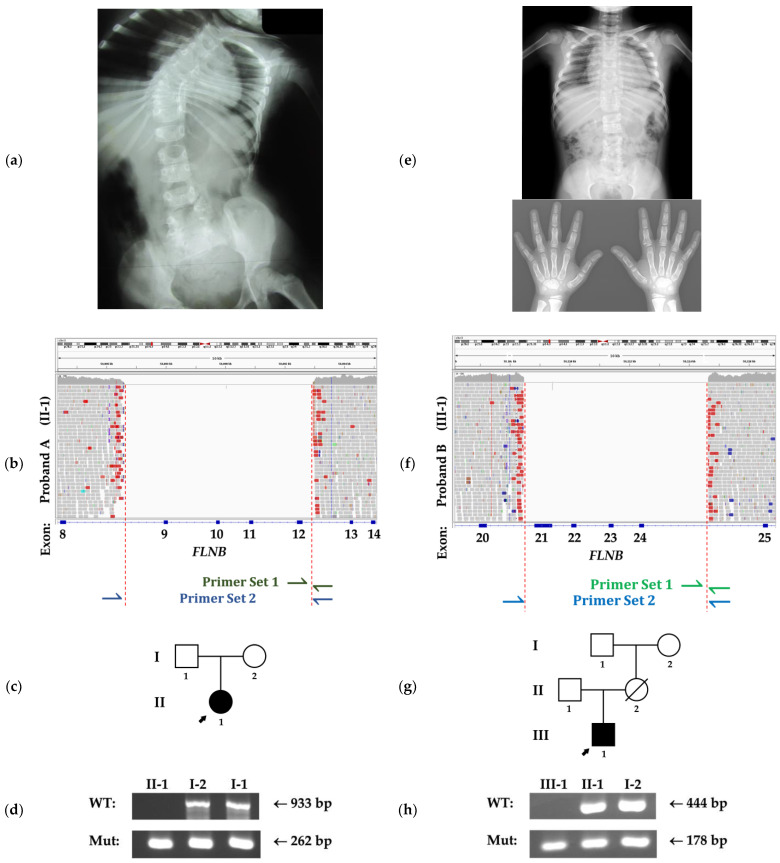
Radiographs and sequence data from families A and B. (**a**) Spine X-ray of proband A at age 6 showing fusions of T3–T11 vertebrae and significant scoliosis. (**b**) IGV plot showing the novel homozygous deletion in proband A that removes exons 9–12 of *FLNB*. The positions of the primers used for the allele-specific PCR are indicated underneath. (**c**) Pedigree of family A. (**d**) PCR results from family A demonstrates the absence of the wild-type (WT) allele and the presence of the deletion-containing allele (Mut) in the proband (II-1) while the parents (I-1, I-2) have both WT and Mut alleles. (**e**) X-rays of proband B at age 9 showing fusions of T6–T9 vertebrae and capitate–hamate fusion bilaterally. (**f**) IGV plot showing the novel homozygous deletion in proband B that removes exons 21-24 of *FLNB*. The positions of the primers used for the allele-specific PCR are indicated underneath. (**g**) Pedigree of family B. (**h**) PCR results from family B showing the absence of the WT allele and the presence of the deletion-containing allele in the proband (III-1), while both alleles are found in the father (II-1) and the maternal grandmother (I-2).

## Data Availability

The identified deletions in *FLNB* were deposited in the ClinVar database under the accession numbers, SCV001482266 and SCV001482267. The data presented in this study are available on request from the corresponding author. The data are not publicly available due to privacy and ethical restrictions.

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
