# Peer review of "Intragenic Deletions in FLNB Are Part of the Mutational Spectrum Causing Spondylocarpotarsal Synostosis Syndrome"

_genes, 2021, doi:10.3390/genes12040528_

Round 1

Reviewer 1 Report

The authors present a well-written report on two cases of spondylocarpotarsal synostosis syndrome caused, and illustrate the added value of copy number sensitive methods. I do, however, have two comments.

  • Heterozygous FLNB pathogenic variants cause different skeletal dysplasias. Did the parents of the affected patients show any signs of skeletal dysplasia? Are different types of mutations reported in e.g. Larson syndrome compared to SCT? The authors should discuss this in the relevant sections of the paper.
  • p.4 line 131 'Both deletions reported here involve the removal of several exons and result in a frameshift'. Have you checked the consequences of the mutations at mRNA level? Or are you able to?

Author Response

Thank you for reviewing our manuscript and providing your comments and suggestions. We have outlined our responses below and hope that these are to your satisfaction.

The authors present a well-written report on two cases of spondylocarpotarsal synostosis syndrome caused, and illustrate the added value of copy number sensitive methods. I do, however, have two comments.

Point 1: Heterozygous FLNB pathogenic variants cause different skeletal dysplasias. Did the parents of the affected patients show any signs of skeletal dysplasia?

Response 1: The parents heterozygous for these identified deletions were normally statured and therefore no radiographs were taken. We have added to the Clinical Reports section that the parents were healthy [lines 88 & 110]

Point 2: Are different types of mutations reported in e.g. Larson syndrome compared to SCT? The authors should discuss this in the relevant sections of the paper.

Response 2: Thank you for this suggestion, we have added into the Introduction: “In contrast, SNVs and in-frame indels that are predicted to have a gain-of-function effect, are responsible for a number of autosomal dominant skeletal dysplasias including Larsen syndrome (MIM# 150250), atelosteogenesis type I (MIM# 108720) and type III (MIM# 108721), and boomerang dysplasia (MIM# 112310)” [lines 39-42]

Point 3: p.4 line 131 'Both deletions reported here involve the removal of several exons and result in a frameshift'. Have you checked the consequences of the mutations at mRNA level? Or are you able to?

Response 3: Unfortunately, live cells or tissues from the families were not available and therefore studies to confirm the effects of the deletions at the transcript level could not be performed. We have acknowledged this in the Discussion [lines 152-153].

Reviewer 2 Report

The authors present two further cases of spondylocarpotarsal synostosis syndrome in which homozygous intragenic deletions in FLNB were identified. As far as we are aware, large structural variants in FLNB have never hitherto been reported in SCT. These cases markedly widen the molecular spectrum of this condition and underline the need for copy-number sensitive methods to be adopted in the search for a genetic diagnosis in patients with clinically diagnosed SCT.

The paper is very well written and clearly structured. It will prove to be useful for clinical geneticists and for molecular geneticists.

Author Response

The authors present two further cases of spondylocarpotarsal synostosis syndrome in which homozygous intragenic deletions in FLNB were identified. As far as we are aware, large structural variants in FLNB have never hitherto been reported in SCT. These cases markedly widen the molecular spectrum of this condition and underline the need for copy-number sensitive methods to be adopted in the search for a genetic diagnosis in patients with clinically diagnosed SCT.

Point 1: The paper is very well written and clearly structured. It will prove to be useful for clinical geneticists and for molecular geneticists.

Response 1: Thank you very much for reviewing our manuscript.

Reviewer 3 Report

The authors identified two homozygous intragenic deletions in FLNB in two unrelated patients with typical phenotypes of SCT. The two deletions remove several consecutive exons and are predicted to cause frameshift and nonsense-mediated mRNA decay. The results broaden the molecular spectrum of SCT and imply the significance of the gene copy number analysis such as MLPA, and new generation sequencing in patients with SCT. In addition, this case report provides further evidence that biallelic loss-of-function variants of FLNB are associated with SCT phenotypes. 

Minor points,

  • Please describe whether the parents are consanguineous or not.
  • The primer sequences and the specific positions would be needed for a better understanding of the allele specific PCR analyses. It would be useful to add figures indicating positions of the primers, exons and the deleted regions.

Author Response

Thank you for reviewing our manuscript and providing your comments and suggestions. We have outlined our responses below and hope that these are to your satisfaction.

The authors identified two homozygous intragenic deletions in FLNB in two unrelated patients with typical phenotypes of SCT. The two deletions remove several consecutive exons and are predicted to cause frameshift and nonsense-mediated mRNA decay. The results broaden the molecular spectrum of SCT and imply the significance of the gene copy number analysis such as MLPA, and new generation sequencing in patients with SCT. In addition, this case report provides further evidence that biallelic loss-of-function variants of FLNB are associated with SCT phenotypes. 

Minor points,

Point 1: Please describe whether the parents are consanguineous or not.

Response 1: Family A was from a population with known high rates of consanguinity but details specific to the parents were not known. We have added this to the Clinical Reports section [lines 88-90].

We have conducted ROH analysis to determine the relatedness of the parents in Family A, but the inbreeding coefficient was found to be rather low, for which we provided possible explanations: “This deletion was situated within a 40 Mb autozygous region (NC_000003.11:g.20957810-61653515), as indicated by ROH analysis. The calculated FROH was a rather unremarkable 0.019, with the majority of this value contributed by the single large segment of ROH around FLNB. Possible alternative explanations for apparent homozygosity of the intragenic FLNB deletion, other than inheritance from a common ancestor were considered, such as a large encompassing deletion inherited from one parent or segmental uniparental isodisomy [16]. However, allele specific PCR…”  [lines 125-131]

Family B had known consanguinity, and ROH analysis supported this. We have added into the results section: “The calculated FROH of the proband was 0.173, suggesting significant relatedness between the parents.” [lines 141-142]

We have added the relevant methods for ROH analaysis [lines 68-72].

Point 2: The primer sequences and the specific positions would be needed for a better understanding of the allele specific PCR analyses. It would be useful to add figures indicating positions of the primers, exons and the deleted regions.

Response 2: Thank you for this suggestion – we have added into Figure 1b and 1f (p.3), arrows to depict the positions of the primers in relation to the exons and the deleted regions. We have added a Supplementary Information document with Table S1 containing the primer sequences and Figure S1 showing the design of the allele specific PCR in the context of the FLNB gene structure.

Reviewer 4 Report

With this case report, the authors seek to expand the genetic spectrum of Spondylocarpotarsal synostosis syndrome to include large deletions in the FLNB gene, and advocate the need to employ appropriate testing methods in order to take these copy-number variants into account during the diagnostic process.

The report is interesting and well written. To the best of my knowledge, no other large deletions within the FLNB gene have been reported in the literature or in disease databases. Therefore, the authors delineate a solid and useful case in the context of SCT diagnostics.

I would just have a couple of minor suggestions for the authors to consider:

  1. they could specify whether they knew or suspected consanguinity for proband A in the Clinical Report section, or whether ROH analysis revealed consanguinity in the Results section (if this was in fact the case);
  2. in the Discussion (perhaps in the last paragraph), they might stress that CMA would not be an appropriate CNV-detection method for deletions within FLNB: this is quite obvious to geneticists, but it may be helpful for some of the other clinicians.

Author Response

Thank you for reviewing our manuscript and providing your comments and suggestions. We have outlined our responses below and hope that these are to your satisfaction.

With this case report, the authors seek to expand the genetic spectrum of Spondylocarpotarsal synostosis syndrome to include large deletions in the FLNB gene, and advocate the need to employ appropriate testing methods in order to take these copy-number variants into account during the diagnostic process. 

The report is interesting and well written. To the best of my knowledge, no other large deletions within the FLNB gene have been reported in the literature or in disease databases. Therefore, the authors delineate a solid and useful case in the context of SCT diagnostics.

I would just have a couple of minor suggestions for the authors to consider:

Point 1: they could specify whether they knew or suspected consanguinity for proband A in the Clinical Report section, or whether ROH analysis revealed consanguinity in the Results section (if this was in fact the case);

Response 1: Family A was from a population with known high rates of consanguinity but details specific to the parents were not known. We have added this to the Clinical Reports section [lines 88-90].

We have conducted ROH analysis to determine the relatedness of the parents in Family A, but the inbreeding coefficient was found to be rather low, for which we provided possible explanations: “This deletion was situated within a 40 Mb autozygous region (NC_000003.11:g.20957810-61653515), as indicated by ROH analysis. The calculated FROH was a rather unremarkable 0.019, with the majority of this value contributed by the single large segment of ROH around FLNB. Possible alternative explanations for apparent homozygosity of the intragenic FLNB deletion, other than inheritance from a common ancestor were considered, such as a large encompassing deletion inherited from one parent or segmental uniparental isodisomy [16]. However, allele specific PCR…”  [lines 125-131]

We have added the relevant methods for ROH analaysis [lines 68-72].

Point 2: in the Discussion (perhaps in the last paragraph), they might stress that CMA would not be an appropriate CNV-detection method for deletions within FLNB: this is quite obvious to geneticists, but it may be helpful for some of the other clinicians.

Response 2: Thank you for this suggestion, we have modified the sentence about CNV-detection methods in the Discussion to include this point: “This risk can be mitigated by using copy number sensitive methods such as WGS or multiplex ligation-dependent probe amplification (MLPA) but not chromosomal microarray as this latter option lacks the resolution to detect all possible intragenic multi-exonic deletions.” [lines 163-164]